# Non-linear enhancement of ultrafast X-ray diffraction through transient resonances

Stephan Kuschel [1,2,3,4] ✉, Phay J. Ho [5] ✉, Andre Al Haddad[5,6], Felix F. Zimmermann [1,2,7], Leonie Flueckiger[8], Matthew R. Ware[1,2], Joseph Duris[2], James P. MacArthur[2], Alberto Lutman [2], Ming-Fu Lin [2], Xiang Li [2,9], Kazutaka Nakahara[2], Jeff W. Aldrich[2], Peter Walter [2], Linda Young [5,10], Christoph Bostedt[5,6,11], Agostino Marinelli [2] ✉ & Tais Gorkhover [1,2,3] ✉

Diffraction-before-destruction imaging with ultrashort X-ray pulses can visualize non-equilibrium processes, such as chemical reactions, with sub-femtosecond precision in the native environment. Here, a nanospecimen diffracts a single X-ray flash before it disintegrates. The sample structure can be reconstructed from the coherent diffraction image (CDI). State-of-the-art X-ray snapshots lack high spatial resolution because of weak diffraction signal. Bleaching effects from photo-ionization significantly restrain image brightness scaling. We find that non-linear transient ion resonances can overcome this barrier if X-ray laser pulses are shorter than in most experiments. We compared snapshots from individual $\approx 100$ nm Xe nanoparticles as a function of pulse duration and incoming X-ray fluence. Our experimental results and Monte Carlo simulations suggest that transient resonances can increase ionic scattering cross sections significantly beyond literature values. This provides a novel avenue towards substantial improvement of the spatial resolution in CDI in combination with sub-femtosecond temporal precision at the nanoscale.

The combination of coherent diffraction imaging (CDI) with bright and focused femtosecond short flashes from X-ray Free Electron lasers (FELs)[1–7] has brought to light transient non-equilibrium phenomena such as metastable stages of metal nanoparticle formation[8], unexpected morphologies diversity in soot formation[9], complex light-matter interaction dynamics[10–16], vortex organization in superfluid droplets[17], relaxation of superheated particles on the femtosecond time scale[12,15,18] and otherwise inaccessible states of water[19]. Theoretical studies suggest that sub-nm spatial resolution within a single FEL exposure is feasible under ideal FEL pulse conditions[1,20–23]. Such spatial resolution in combination with sub-fs temporal precision would

transform our capabilities to follow ultrafast phase transitions, photochemical or catalytic reactions at the single nanoparticle level in native state. However, state-of-the-art spatial resolution in experiments on individual heavy element nanoparticles is limited to a few nanometers even in the hard X-ray regime[7,23–25]. Further increase in spatial resolution requires an enhancement in diffraction image brightness. In the linear semiclassical model, the scattered photon number from a specimen is proportional to the incoming X-ray photon fluence $F_{ph}$ (incoming photons per area) and a constant, wavelength-specific elastic scattering cross section $\sigma_{scat}$ of the material[26–28]. This assumption breaks down for high FEL fluences when the image brightness is

[1]SLAC National Accelerator Laboratory, Stanford PULSE Institute, Menlo Park, CA, USA. [2]SLAC National Accelerator Laboratory, Menlo Park, CA, USA. [3]Institute for Experimental Physics/CFEL, University of Hamburg, Hamburg, Germany. [4]Technical University Darmstadt, Institute of nuclear physics, Darmstadt, Germany. [5]Chemical Sciences and Engineering Division, Argonne National Laboratory, Lemont, IL, USA. [6]Paul Scherrer Institute, Villigen, Switzerland. [7]IOAP, Technical University of Berlin, Berlin, Germany. [8]La Trobe University, Latrobe, Australia. [9]Department of Physics, Kansas State University, J.R. Macdonald Laboratory, Manhattan, KS, USA. [10]Department of Physics and James Franck Institute, The University of Chicago, Chicago, IL, USA. [11]LUXS Laboratory for Ultrafast X-ray Sciences, Institute of Chemical Sciences and Engineering, Lausanne, Switzerland. ✉e-mail: stephan.kuschel@tu-darmstadt.de; pho@anl.gov; marinelli@slac.stanford.edu; tais.gorkhover@cfel.de

affected by bleaching effects inside the sample, which reduce the effective scattering cross section during the FEL exposure[14,20,21,23,29–35].

Bleaching is a result of a complex interplay between electronic and ionic structural damage of the sample interacting with an intense X-ray FEL pulse. Electronic damage is initiated by photoionisation processes, which are inevitable when the specimen is exposed to intense X-ray pulses. Sequential multi-photon absorption and subsequent relaxation processes such as Auger/Coster-Kronig decays remove the electrons from the parent ions on sub-fs to 10-fs time scale. Ionization increases the transparency of the sample as a shrinking number of bound electrons decreases the absorption/scattering cross sections of an ion according to the semi-classical model[21,30,31]. The subsequent damage to the ionic structure through hydrodynamic expansion is driven by the hot electrons trapped by a rapid build up of space charge[12,36]. There is a consensus that the FEL pulse duration must be shorter than the anticipated onset of ionic structure damage. But electronic structure modification cannot be avoided during the FEL exposure. Thus, most studies regard electronic bleaching through

photoionisation as an inevitable limit to image brightness scaling with $F_{ph}$, which is detrimental to the quality of the diffraction images[21,23,24,29,31,32,37].

In contrast to previous studies, in this work we find a more complex picture where increased $F_{ph}$ leads to higher photoionisation rates but also to enhancement of the effective scattering cross section if FEL pulses are extremely short and tailored to excite multiple transient ion resonances. In our study, we explore the brightness and spatial resolution of diffraction images of individual Xe nanoparticles as a function of FEL pulse parameters such as FEL photon energy $h\nu$, pulse energy $E_p$, and duration $\tau$ (all parameters are summarized in Supplementary Table 1). We scanned the photon energy $h\nu$ in the vicinity of the Xe 3d absorption edge.

## Results

### Non-linear X-ray scattering response in single nanoparticles

A detailed schematic of our experiment is shown in Fig. 1a. Individual near-spherical Xe nanoparticles with diameters of 60–150 nm intersect

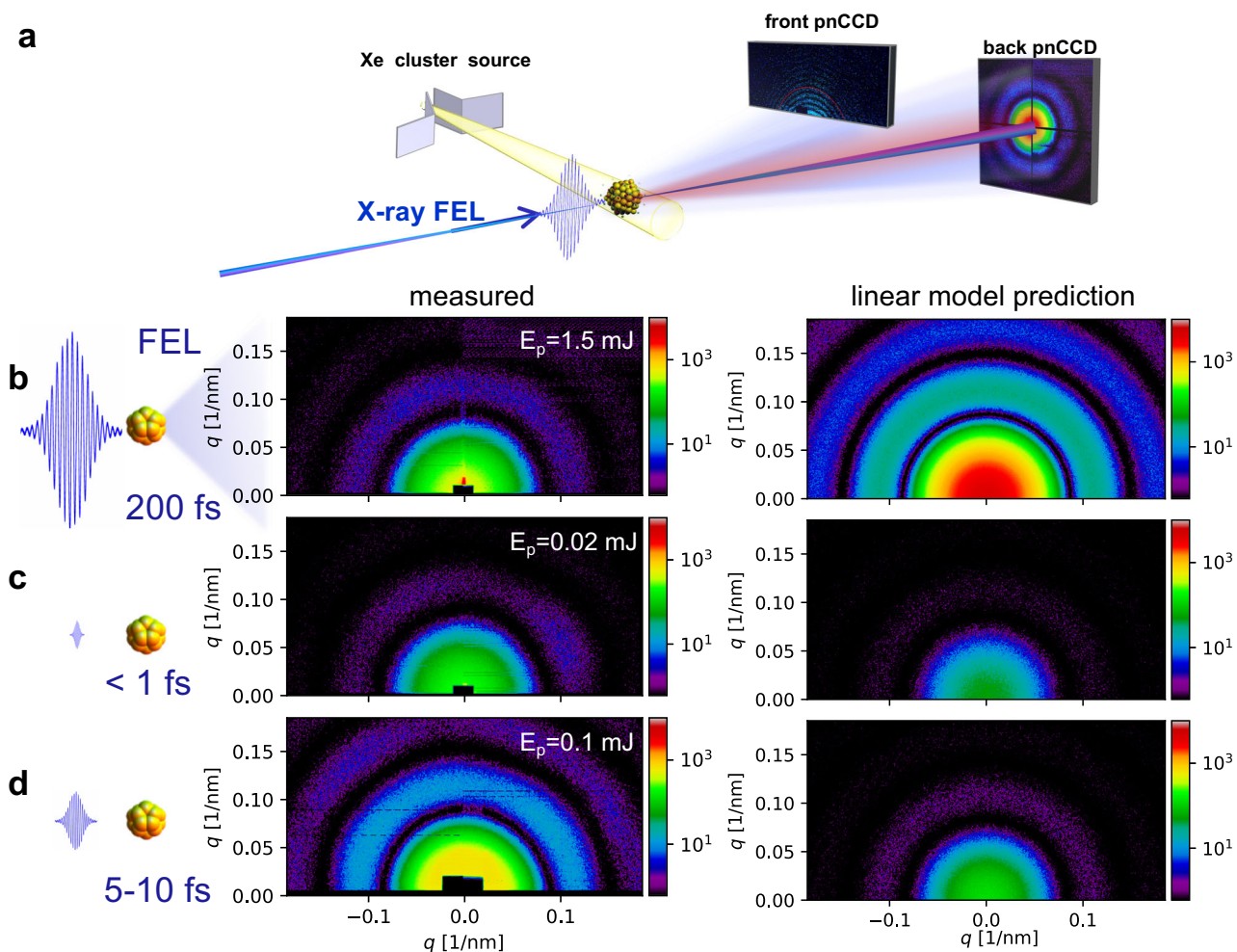

**Fig. 1 | The schematic of the experimental setup and brightest images. a** Intense X-ray FEL pulses were focused and intersected with a stream of Xe clusters. The resulting single-shot, single-particle X-ray diffraction patterns were recorded using pnCCD detectors[38,40] positioned further downstream of the interaction point. The front pnCCD is positioned at a distance of 130 mm and the rear pnCCD at 732 mm from the X-ray focus. The brightest X-ray diffraction patterns of individual Xe nanoparticles with diameters 97 ± 3 nm are displayed for **b** 200 fs, **c** 5–10 fs, and **d** sub-fs pulse durations on the left side. The x and y axis of images recorded by the rear pnCCD are plotted in q wave vector space. The color scale indicates detected photons per pixel. The sub-fs image was recorded with a $E_p = 0.02$ mJ pulse energy at $h\nu = 735$ eV X-ray photon energy, the 200 fs with $E_p = 1.5$ mJ at $h\nu = 730$ eV, the 5 fs

with $E_p = 0.1$ mJ pulse energy at $h\nu = 1500$ eV. The brightest image was not recorded using the strongest FEL pulse. The diffraction images expected from the semi-classical picture are plotted on the right side of (**b**–**d**). The right side images are calculated by Condor software[45] using the experimental FEL fluence and the quantum efficiencies of the pnCCD detector. According to linear scaling with incoming X-ray pulse energy one expects that the brightest X-ray diffraction snapshots should be observed from 1.5 mJ, 200 fs FEL pulses ((**b**), dark blue). In contrast, we find that the diffraction patterns which were recorded with only 0.1 mJ and 5 fs short pulses are the brightest. We attribute the increased brightness to transient resonances.

the path of focused and intense single X-ray FEL pulses inside the LAMP endstation at the Linac Coherent Light Source (LCLS)[38,39]. Single-particle, single-exposure X-ray diffraction snapshots are recorded at 30 Hz using p-n junction charge-coupled (pnCCD) devices[40] located at two different positions further downstream from the interaction region (see Supplementary Note 3). The particle size is directly encoded into the Airy pattern-like diffraction patterns and can be recovered with ± 0.3 nm precision (see Supplementary Note 4). We scanned the photon energy $hv$ in the vicinity of the Xe 3d absorption edge between 650 eV < $hv$ < 740 eV. In addition, we recorded images at $hv$ = 1500 eV where transient resonances (TRs) in Xe gas have been observed previously[41]. Each scanning step contains thousands of diffraction patterns from single Xe nanoparticles with fluctuating brightness due to random positions inside the 1.5 μm FEL focus with a near Gaussian X-ray fluence distribution[12]. We have investigated three FEL pulse durations: >100 fs as a pulse duration found in many ultrafast soft X-ray CDI studies[4,6,9,23,42,43], 5–10 fs as a pulse duration beating the onset of the ionic structure damage of the specimen, and newly available sub-fs pulses[44] outrunning significant parts of electronic damage through relaxation processes such as Auger decay (Xe M-shell Auger lifetime ≈ 1 fs).

The comparison of the brightest images for each pulse duration sheds light on the scattering efficiency at different time scales. Some of the brightest single X-ray diffraction snapshots per FEL pulse duration are displayed on the left side of Fig. 1b–d. The diffraction patterns stem from similarly sized nanoparticles with diameters 97 ± 3 nm. Intuitively, one would expect that the brightest images are recorded using a 200 fs pulse with the highest pulse energy $E_p$ = 1.5 mJ, which corresponds to the highest fluence $F_{ph}$ (Fig. 1b). In fact, the most intense X-ray diffraction snapshots were taken with few femtosecond pulses and only $E_p$ = 0.1 mJ (Fig. 1d).

The strong non-linearity of the measured scattering process emerges from a comparison between the experimental data and the simulation of the anticipated images based on the linear semi-classical model[45]. Here we assume that the brightest images originate from the vicinity of the FEL focus center with $F_{ph}^{max}$. The right-hand side images in Fig. 1b–d display extrapolated brightest shots based on linear scaling with $F_{ph}^{max}$ and literature values of $\sigma_{scat}$. The values for $F_{ph}^{max}$ are deduced from beam line parameters and statistical data analysis, Supplementary Note 3. The images recorded with sub-fs and few fs FEL pulses appear brighter than linear scaling with $F_{ph}^{max}$ would predict. In contrast, the experimental diffraction patterns recorded with 200 fs and highest $F_{ph}^{max}$ are one order of magnitude dimer.

This observation points to a complex interplay between electronic and ionic damage occurring on different time scales. The dim 200 fs images could be explained by typical bleaching and ion structural damage observed in previous studies. However, the brightness enhancement for 5 fs and sub-fs exposures hints at non-linear effects such as transient resonances (TRs) which can emerge during the FEL exposure.

## Theoretical model of transient resonances (TRs)

TRs arise when core-hole ionization events drive the parent ion into short-lived resonances[10,11,13,37,41,46]. Previous studies in rare gas atoms reported increases in absorption cross-section by almost two orders of magnitude due to individual TRs in Ne[46] and even TRs cascades in Xe atoms at 1500 eV[41]. The potential of TRs for imaging has received little attention so far as first imaging experiments with 100–200 fs long pulses on single nanoparticles suggested rather detrimental effects[10,11,13,37].

The TR mechanism is illustrated in Fig. 2 based on the mechanism of 3d → 4f resonant X-ray scattering in a neutral Xe atom and a core-excited Xe+* ion. Resonant scattering occurs through virtual electron transitions between the 3d core level and the unoccupied 4f state as illustrated for neutral Xe (Fig. 2a). If X-rays are scattered by the core-

hole excited Xe+* ion with a modified Coulomb potential (Fig. 2b), the binding energy of the excited 3d* orbital shifts from $hv_1$ (black dashed arrows) to $hv_2^* \approx hv_1 + 40$ eV (red arrows). In addition, the 4f* orbital in Xe+* is pulled closer towards the core in order to screen the positive charge, as visualized by the absolute value of the wave function $|\psi|$ (Fig. 2b, top graph). This orbital restructuring increases the overlap between the 3d* and 4f* orbitals and thus, the transition dipole moments. The resulting augmented 3d* → 4f* scattering cross section $\sigma_{scat}^*$ is more than one order of magnitude higher than the neutral Xe $\sigma_{scat}$ (red vs black dotted line in Fig. 2c, more details in Supplementary Note 1).

The overall lifetime of a single TR is limited by the core-hole decay time and by the photoionisation rate. However, multiple TRs pathways can appear as the FEL pulse excites a range of ionization states. A previous absorption spectroscopy study on FEL ionized Xe atoms suggested that sequential multi-photon absorption can trigger TR cascades throughout the entire FEL exposure duration at $hv \approx 1500$ eV[41]. Such TR cascades arise if more than one photon per atom is absorbed. During the FEL pulse exposure in the soft X-ray region, the most likely photo-ionization process is the removal of a single core M-shell electron. Depending on the FEL fluence and photon energy, subsequent core holes can be created through sequential photo-ionization. Such highly excited ions usually recombine through Auger decay and open new vacancies in the valence shells within femtoseconds. Nanoplasma effects such as collisional and frustrated ionization can further promote the removal of outer shell electrons[10–12].

## Sub-10 femtosecond short FEL pulses benefit the image quality

Such TR cascades at photon energies above classical absorption edges provide a new dimension to counter and even overcompensate bleaching effects if the FEL parameters are chosen appropriately. An important aspect is that an increase $\sigma_{scat}^*$ often comes at the cost of enhanced absorption cross-sections of ions because both cross-sections are intertwined through the Kramers–Kronig relations. Stronger absorption cross sections ultimately mean that ionic structure damage is accelerated and thus, FEL pulse durations considered standard for imaging might result in degraded images if the FEL pulses are very intense. A statistical analysis of the scattering cross-sections $\sigma_{scat}^{exp}$ extracted from thousands of recorded images supports the hypothesis that resonant X-ray diffraction from excited ions can be both, detrimental or beneficial to the image brightness and quality. Our data overview illuminates the full extent of bleaching in 200-fs snapshots and the enhancement of $\sigma_{scat}^{exp}$ in diffraction patterns recorded with sub-fs and few-fs pulses as summarized in Fig. 3a–c. We calculated $\sigma_{scat}^{exp}$ per Xe atom for the top 5% brightest images for each incoming X-ray photon energy scan step. Each dot represents a single measured diffraction image of an individual Xe nanoparticle that is sufficiently bright for an automated nanoparticle size fitting routine (see Supplementary Note 4). The individual value for $\sigma_{scat}^{exp}$ for the top 5% shots can be directly deduced from the image brightness corrected for the exposure fluence $F_{ph}$ and the nanoparticle size (see Supplementary Note 5). The absolute value for $F_{ph}$ has been cross-checked using a calibration correlating the measured M-shell fluorescence yield (see Supplementary Note 8), and the pulse energy values reported by the gas detector (see Supplementary Note 3).

The corresponding literature values for $\sigma_{scat}$ for the neutral Xe atom are visualized by the dashed gray line[47]. For all pulse durations, $\sigma_{scat}^{exp}$ values follow closely the literature Xe values near the M shell 3d resonance ($hv$ < 720 eV) with the exception of most intense 200 fs pulses. The situation changes above the M-edge. First, for all three pulse durations, conditions above the M-shell exist when $\sigma_{scat}^{exp}$ significantly exceeds the Xe literature value for the corresponding photon energy above 720 eV (see arrows in Fig. 3a–c). In all $\sigma_{scat}^{exp}$ enhancement cases, coincident measurements of fluorescence yield confirm that on

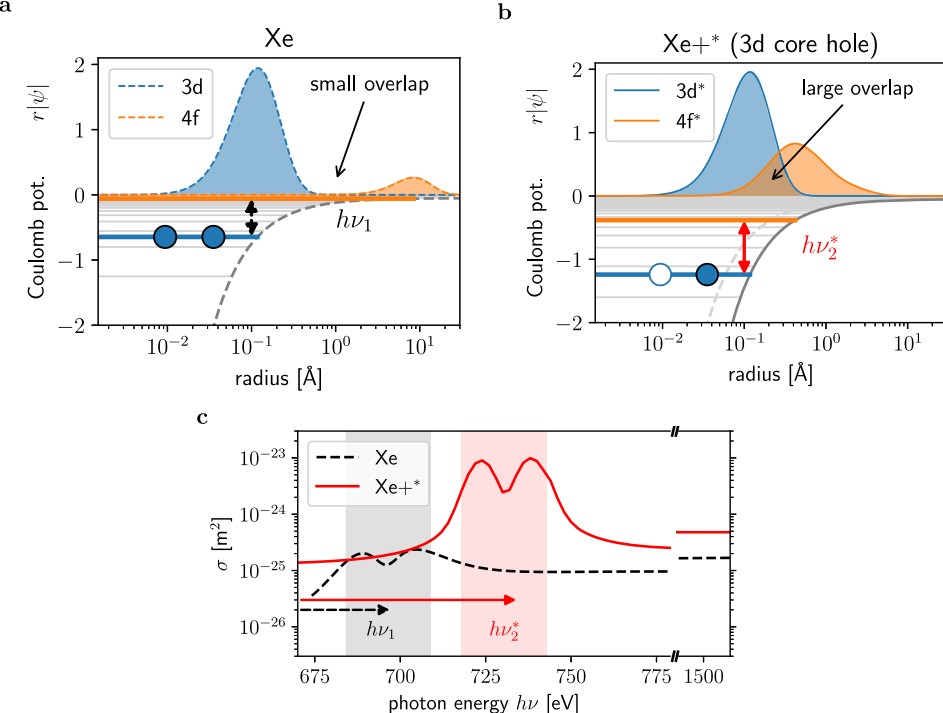

**Fig. 2 | The concept of transient resonances.** In **a**, the simulated Coulomb potential is plotted versus the distance from the charged core for a neutral Xe atom (gray dashed line) and in **b** for a core excited Xe+* (gray solid line). The corresponding 3d (blue) and 4f (orange) orbitals are displayed for a neutral Xe atom (**a**), dashed lines) and 3d* and 4f* excited Xe ion with a core-hole (**b**), solid lines). First, the core-hole excitation shifts the binding energy of the remaining 3d* electrons from $h\nu_1$ (black) to greater $h\nu_2^*$ (red) due to a modified Coulomb potential. Second, the core-hole excitation also rearranges the orbitals to 3d* (blue) and 4f* (orange) plotted on top of the Coulomb potential. Due to the core-hole induced charge imbalance the 4f* orbital is pulled closer to the atom's center by almost two orders of magnitude creating a strongly increased overlap with the 3d* wave function and hence an increased transition dipole strength for the 3d* → 4f* transition. This has dramatic consequences for the scattering cross-section in the vicinity of the Xe 3d absorption edge. In **c**, the scattering cross section $\sigma_{scat}$ for the neutral Xe (dashed black line) and $\sigma_{scat}^*$ for the excited Xe+* (red solid line) is plotted vs the incoming X-ray photon energy. The excited atom Xe+* scatters two orders more strongly than neutral Xe at the shifted resonance position $h\nu_2^*$ and almost two orders more compared to the neutral Xe resonance maximum $h\nu_1$. Energies, cross-sections, and radial wave functions displayed here were calculated using Hartree-Fock-Slater simulations (see Supplementary Note 1). The gray lines represent a schematic drawing of the Coulomb potential. The shape of the wave functions is a result of the Hartree–Fock simulations and its integral is normalized to one.

average more than one X-ray photon per atom has been absorbed (see Supplementary Note 8) For short pulses, brightness of the images can increase non-linearly with incoming fluence. At $h\nu = 1500$ eV and 5-10 fs pulse duration (red dots), the enhancement is -10 times higher than any point on the dashed Xe literature value curve and also in comparison with weaker FEL pulses (blue dots). Second, for longer pulses the role of TRs is more ambiguous. The $\sigma_{scat}^{exp}$ extracted from images recorded with 200-fs and $E_p = 1.5$ mJ pulses (blue dots) is overall 10 times lower than Xe literature values. When 200 fs pulses are attenuated, $\sigma_{scat}^{exp}$ is augmented for photon energies above the M-shell.

Our simulated $\sigma_{scat}^{*29}$ (solid lines in Fig. 3a–c) is overall in good agreement with corresponding $\sigma_{scat}^{exp}$. For 200 fs FEL exposures with $E_p = 1.5$ mJ pulses, our calculation predicts a significant reduction of the scattering cross section due to the rapid disintegration of the ion structure fueled by TRs (see Supplementary Note 2 and Supplementary Fig. 1). The presence of TRs above the 3d M-shell absorption edge for shorter and/or longer but weaker pulses is also supported by our calculation. If the sample is exposed to attenuated 200 fs pulses, the hydrodynamic expansion occurs much slower, and thus, the scattering enhancement of TRs reappear. Interestingly, the simulation underestimates the non-linear scattering enhancement at $h\nu = 1500$ eV. One possible reason is that our simulation neglects ionization potential suppression[48,49] which can rapidly delocalize the valence electron shells and lead to extremely efficient TR cascades[36]. Assuming that Xe $n = 5$ and $n = 4$ shells are fully removed, our simulation predicts $\sigma_{scat}^*$ up to 9 times the neutral Xe cross section for 5–10 fs FEL pulses at $h\nu = 1500$ eV in agreement with our data. An indirect support for a significant role of

complex nanoplasma ionization effects stems from the coincident extreme fluorescence yield (see Supplementary Note 8).

A cascade of multiple TRs contributes to the total enhancement of the scattering cross-section over the duration of the incident X-ray pulse. Above the M-shell absorption edge at $h\nu = 730$ eV, $\sigma_{scat}^*$ of Xe ions with charge states 1+-20+ is at least one order of magnitude greater than $\sigma_{scat}$ as demonstrated in Fig. 4a. If more than one X-ray photon per Xe atom is absorbed on average, the Xe ions inside the nanoparticles will run through a series of excitation and relaxation channels which will contribute to the enhancement of the integrated $\sigma_{scat}^*$ throughout the FEL exposure. The relaxation processes will reduce the number of core holes but increase the valence vacant states which also exhibit enhanced $\sigma_{scat}^*$. Far above the Xe M-edge at 1500 eV brighter images can be expected if the FEL pulses can produce double-core holes (Fig. 4b). The $\sigma_{scat}^*$ of a double-core Xe ion scatters over 10 times more efficiently than $\sigma_{scat}$. After a few femtoseconds, such core holes can relax and produce a highly charged nanoplasma with valence vacancies 20+ and higher[15,41] which will drive the system into relaxation/excitation cascades with enhanced $\sigma_{scat}^*$.

The increase in the scattering cross section for short pulses directly translates into higher resolution as pointed out in Fig. 3d. The radial plot of one of the brightest diffraction patterns from a single Xe cluster (red line) maintains a high signal-to-noise ratio down to 5–6 nm full-period resolution (Supplementary Note 6). This is comparable to the best spatial resolution achieved so far in single-shot X-ray imaging with higher-Z Au nanoparticles using hard X-rays with one order of magnitude shorter wavelengths[24].

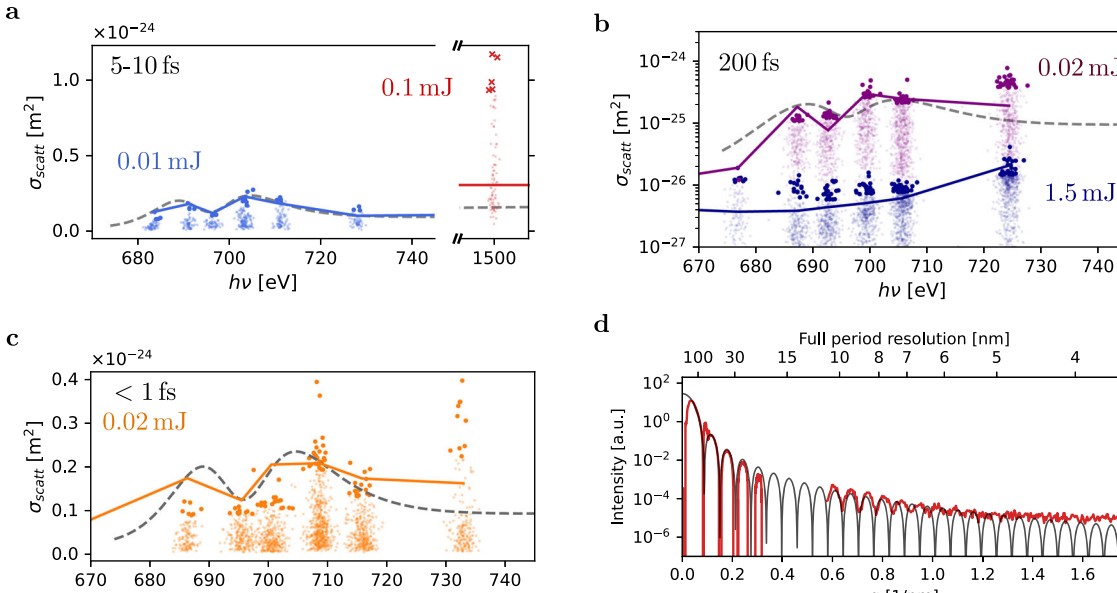

**Fig. 3 | Imprints of transient resonances.** **a**–**c** depict the measured $\sigma_{scat}^{exp}$ in dependence to the incoming X-ray photon energy $h\nu$ for all pulse duration measurements such as 5–10 fs (**a**), 200 fs (**b**), and 0.5 fs (**c**). Each dot in the graph represents a single Xe nanoparticle observed by a single FEL exposure. The point spread stems from the fact that the nanoparticle diameters 60–150 nm are much smaller than the FEL focus average full-width half maximum (FWHM) 1.5 μm. Each randomly injected sample experiences a different exposure FEL fluence inside the FEL focus and only the top few percent of all hits with the highest fluorescence/scattering yields must have been recorded near the FEL focus center with the highest intensities. Thus, $\sigma_{scat}^{exp}$ has been calculated only for the top 5% of all shots as indicated by bold dots (see Supplementary Note 5 and Supplementary Note 8). The gray dashed line indicates the neutral Xe scattering cross section $\sigma$ for comparison. Our calculations (solid lines) agree overall well with the experiment. The strongest enhancement can be found at 1500 eV (red) where TR cascades have been observed in the past in Xe atoms[41]. Above the M-edge, sub-fs pulses are scattered up to 4 times stronger than the linear model predicts. For further discussion see text. **d** demonstrates that increased brightness translates to higher spatial resolution. Here, the radial plot (in red) of one of the brightest diffraction images recorded at 1500 eV is compared to theoretical diffraction signal from a solid sphere (black line). Diffraction signals down to 5–6 nm full period resolution are clearly visible and the overall radial profile follows closely the theoretical calculation for diffraction from a solid sphere[10,13,31]. See Supplementary Note 6.

## Simulation of ideal FEL parameters

Simulations including TRs will improve the outcome of imaging experiments by mapping the best imaging conditions for specific samples. The important parameters include X-ray FEL fluence, FEL pulse duration, and FEL photon energy. The ideal parameters also depend on the desired spatial resolution and the sample size. The parameter space is quite extensive and simulations for each data point require significant computational effort. However, the most important parameters can be estimated without time-dependent simulations.

The choice of the pulse duration depends mainly on the sample size and required spatial resolution. Extrapolating from Supplementary Fig. 1, 10–20 fs will be sufficiently short to images larger samples greater than 60 nm, especially if the interesting structures are inside the sample volume. Such relatively large specimen will undergo a hydrodynamic expansion, where the structural damage will occur first on the outer surface[12,15]. In the case of molecules or clusters smaller than 10 nm, few fs down to sub-fs pulses should be considered. Here, a Coulomb explosion will lead to a rapid disintegration of the specimen if the FEL pulse ionizes more than one atom on average[50]. Sub-fs to few fs pulses will provide advantages when aiming towards atomic resolution, as such short pulses outrun Auger decay and have the potential to deliver snapshots of an almost pristine state of the sample[29].

The choice of the photon energy can be estimated from resonance maps such as depicted in Fig. 4a, b for Xe atoms. Near the M-shell resonance around 700 eV (a), there are clusters/cascades of TRs near single and double core hole states up to Xe 21+ ions. At 1500 eV (b), the TRs can be reached mainly from charge states greater than Xe 21+. Higher FEL fluences will create higher charge states in general, and thus the optimal photon energy will shift to higher photon energies/shorter X-ray wavelengths.

The choice of the FEL fluence is currently limited by the focusing optics available at FEL facilities. A more complex study of optimal FEL parameters for Xe samples including different fluences is shown in Fig. 4c for 10 fs pulses and Fig. 4d for sub-fs pulses. We assume that the FEL is focused to a 1 μm FWHM spot (a value achievable at many end stations), and the fluence can be changed through FEL pulse energy E. In Fig. 4, the photon energy is plotted versus FEL pulse energy E, the color scale indicates the transient scattering cross-section enhancement compared to the literature value of the Xe scattering cross-section. The red area indicates the highest experimentally achievable pulse energies at the end station where the experiment was performed (TMO at the LCLS).

In order to get the highest spatial resolution one would choose the shortest X-ray wavelength in a region where significant TRs can be expected for the experimentally achievable highest X-ray fluence. Based on this simulation, photon energies around 1 keV could in principle deliver almost wavelength-diffraction-limited images. With better optics or modifications of the accelerator, one could reach the 1–10 mJ per square micron regime, where TRs clearly overcompensate electronic bleaching by two orders of magnitude.

## Discussion

Overall, our study illuminates the potential of TRs in ultrafast X-ray CDI. We demonstrate that TRs can significantly increase the brightness of X-ray diffraction images if FEL pulses are shorter than the onset of the ionic structure damage. We observe the enhancement effect across at least one order of magnitude in FEL fluence for pulse durations <10 fs. This data and our simulations indicate that the effect is scalable with FEL pulse energy as subsequent photoionisation events can drive the ions into a resonant cascade similar to REXMI[41] and some resonances enhance diffraction by factors up to

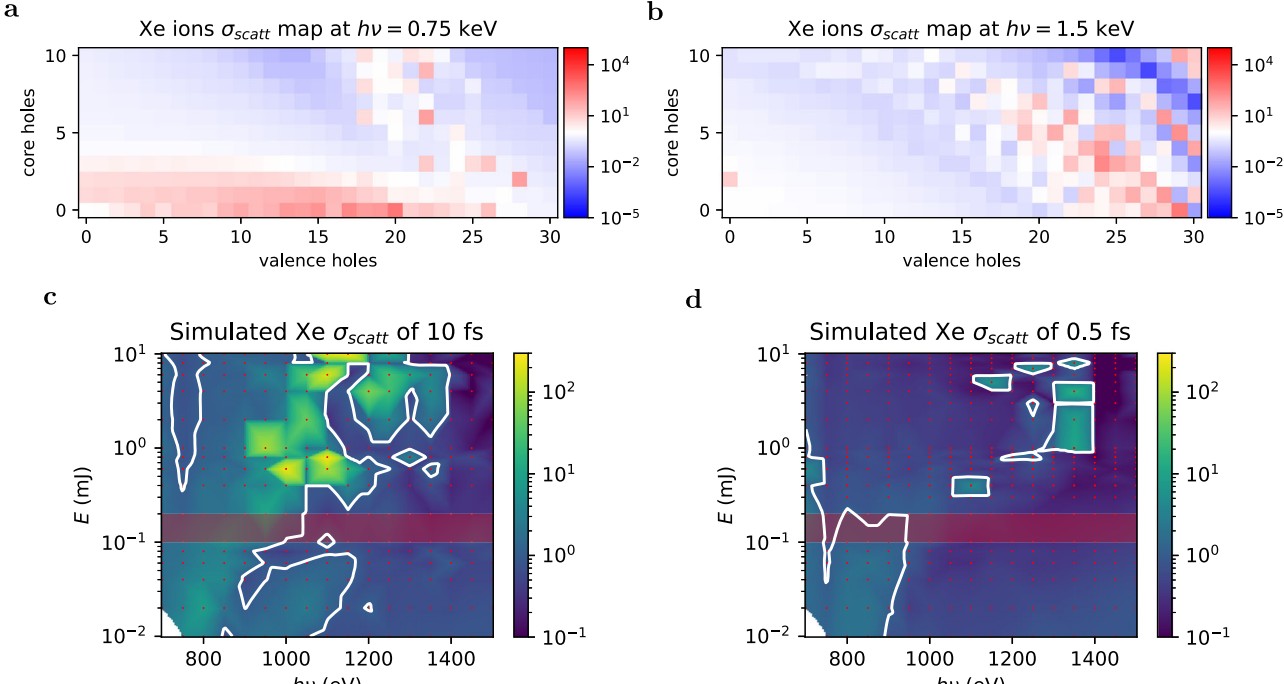

**Fig. 4 | Simulation results.** Upper panels: resonance maps for specific electronic states of Xe ions are depicted in (**a**, **b**). For a number of removed valence electrons (*x*-axis) and a number of removed inner shell 3d M-shell electrons (*y*-axis), the color scale displays the scattering cross-section normalized by the neutral Xe cross-section. At a photon energy $h\nu = 735$ eV **a** enhancement already sets in for low charge states, while at 1500 eV **b** higher charge states are required for enhancement. The simulations suggest that a single charge state can scatter up to a factor $10^4$ more than the neutral Xenon at 1500 eV. Such resonance maps can be used to calculate transient scattering cross-sections for different FEL pulse conditions. Lower panels: examples for such calculations are shown in (**c**, **d**), where the photon energy (*x*-axis) is plotted vs incoming FEL fluence. The fluence is defined through the FEL pulse energy (*y*-axis) focused onto a 1 μm focal spot size. The color scale is the transient Xe ion scattering cross section normalized by the neutral scattering cross section of Xe, the white color corresponds to the neutral Xe atom scattering cross-section. The red dots mark the simulated parameters. The calculation of 10 fs short FEL pulses **c** shows up to 200 times brightness increase in the region of 1000 eV, the equivalent simulation of sub-fs pulses **d** shows less amplification at similar energies as such short pulses outrun relaxation cascades. The state-of-art FEL performance is marked by the red shaded area.

>$10^4$ (see Fig. 4a, b). TRs are not exclusive to Xe, previous spectroscopic studies have found short-lived resonances in oxygen[37] and neon[46]. TRs can be expected in any multi-shell element. In the soft X-ray regime TRs could be combined with super-resolution algorithms[51] to achieve high material contrast from light elements, such as carbon, oxygen, and nitrogen, and high spatial resolution down to single molecule level. In principle, TRs should also exist in the hard X-ray regime, which offers a direct route to Ångstrom resolution in single-shot X-ray imaging of fragile or short-lived states of matter. This would open a plethora of applications such as observation of non-equilibrium processes under ambient conditions at the single nanoparticle level with sub-fs temporal precision. Such capabilities would benefit our understanding of photochemistry, catalysis, and material studies. Beyond that, TRs are element-specific and thus could increase material contrast in X-ray chemical femtosecond crystallography[52] and potentially eliminate the need of heavy stain atoms in time-resolved multi-wavelength anomalous crystallography studies. Our study demonstrates that optimal conditions for diffraction enhancement can be identified via simplified static resonance maps or more complex time-resolved simulations as described in Fig. 4. In principle, TRs can be excited by an optical pre-pulse and significantly increase signal-to-noise ratio in state-of-the-art ultrafast table-top XUV imaging setups[53].

Our study emphasizes the significance of intense and <10 fs FEL pulses, which have received little attention so far in CDI. During the time of the present study, the FEL pulse energies for very short pulses were moderate and could not be easily scaled. Current developments in accelerator science are already changing the availability of very intense few fs and sub-fs pulses[44,54–56]. Newly available intense sub-fs

FEL pulses can become a significant driver for ultrafast CDI as relaxation processes such as Auger decays of K- and L-shells happen mostly within hundreds of attoseconds. Our results pave the way for improvement of ultrafast CDI based on non-linear effects, which has the potential to combine atomic resolution with attosecond temporal precision. This will open new avenues for studies of non-equilibrium states of matter in chemistry and material science.

## Data availability
The brightest images mentioned in this study have been deposited in a git repository accessible at https://github.com/skuschel/XenonNanoparticlesAgoSecondPulses. The raw data is available upon request.

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

## Acknowledgements

Use of the Linac Coherent Light Source (LCLS), SLAC National Accel-erator Laboratory, is supported by the U.S. Department of Energy (DOE), Office of Science, Office of Basic Energy Sciences (BES) under Contract No. DE-AC02-76SF00515. S.K. and T.G. were supported by the U.S. DOE BES Chemical Sciences, Geosciences and Biosciences Division through the Panofsky fellowship from SLAC National Laboratory. P.J.H. and L.Y.

were supported by the U.S. DOE BES Chemical Sciences, Geosciences, and Biosciences Division under Contract No. DE-AC02- 06CH11357. F.F.Z. and M.R.W were supported by the U.S. DOE BES Chemical Sciences, Geosciences and Biosciences Division, Chemical Sciences, Geosciences and Biosciences Division through the Early Career Research Program project number 100482. We acknowledge financial support from the Open Access Publication Fund of Universität Hamburg.

## Author contributions

T.G. and A.M. conceived the idea for the experiment based on discussions with C.B. J.D., J.P.MA., A.L., and A.M. developed and operated the XLEAP sub-fs mode. M.-F.L., X.L., K.N., J.W.A., and P.W. prepared the beamline for the experiment. The experimental setup was planned and performed by all authors. The experiment was performed by S.K., A.A.H, F.F.Z, L.F., M.R.W, J.D., J.P.MA., A.L., M.-F.L., X.L., K.N., J.W.A, P.W., L.Y., C.B., A.M., T.G., and S.K. led the data analysis. P.J.H. performed the simulations. S.K., P.J.H., C.B., and T.G. wrote the manuscript with input from all authors. All authors reviewed the manuscript.

## Funding

## Competing interests

The authors declare no competing interests.
