## [Transparent Peer Review file · Nature Communications]

Non-linear enhancement of ultrafast X-ray diffraction through transient resonances

Corresponding Author: Professor Tais Gorkhover

Version 0:

Reviewer comments:

Reviewer #1

(Remarks to the Author)

The manuscript by S.Kuschel, et al. reports new investigation addressing the core-level ionization induced strong radial contraction of the valence orbitals with sensitive dependence on the XFEL pulse duration. Importantly such alteration gives a direct impact on the X-ray scattering cross section of the target element, which can be employed as a clever route to enhance diffraction signals. The diffraction signal enhancement is attained from the increase in radial overlap integral to effectively accommodate larger scattering cross section. Overall, the manuscript is well designed by focusing on this point clearly with convincing demonstration combining both experimental data and simulation results. The influence of the XFEL pulse duration on the diffraction signal had been pointed out while ago, which could not receive good attentional without experimental evidence. The presented research provides a clear and impactful advance now to demonstrate it in real experiments. As such, I think the manuscript worth publication in the current journal. I add the following points to be addressed before its publication to make the manuscript scientifically solid.

1. The message becomes unclear if readers want to gain idea about what the sensitive parameter is and how to reach optimum condition. The reason comes from the structure of the manuscript dealing with three parameters of laser fluence, pulse duration, and pulse energy, but without systematic description/investigation. For instance, the pulse duration shorter than AA fs is recommend based on BB analysis. Pulse intensity dependence for each pulse duration needs to be compared for the diffraction signal. And finally how the photon energy, with respect to the absorption edge (or core-level binding energy), shows the dependence on the strength of the diffraction signal? All these make multi-dimensional parameter analysis, which is essential for reaching optimal experimental condition as the manuscript's main message points to. I understand providing multi-dimensional analysis data from experiments is not amenable, but at least theoretical investigation has to be provided systematically.
2. In Fig.2, please state clearly which one is cartoon picture, and which ones are real simulation results. How was the gray dotted and solid lines are obtained?
3. Explicit relation between the scattering cross section and overlap integral needs to be stated in the manuscript.
4. It is also important to state about the quality of the diffraction signals themselves in addition to the signal strength only. The authors should add enough investigation results on this. For instance, any occurrence of distorted diffraction signal with fluence and pulse duration dependence?

Reviewer #2

(Remarks to the Author)

The article "Non-linear enhancement of ultrafast X-ray diffraction through transient resonances" by Kuschel et al. presents an interesting application of single pulse x-ray free electron laser radiation to study Xe nanoclusters. The authors illuminated nanoclusters with single XFEL pulses of various pulse lengths and show compelling evidence that transient resonances induced by the highly ionized states greatly increase the scattering efficiency. The results seem valid and compelling, the presentation is clear and concise. However, the current state of the paper does not seem sufficiently compelling to merit publication in Nature Communications unless the following questions could be addressed.

Major concerns/questions:

- 1) Does the current technique have any applicability outside of Xe nanoclusters? Can this transient resonance impact, for instance, biological cell imaging or serial femtosecond crystallography? What evidence is there of this being applicable to other areas of XFEL or x-ray science beyond Xe nanoclusters? This argument to me seems to greatly affect whether the impact of this work is sufficiently broad for a wider audience such as Nature Communications readers, or whether this work should appear in a more focused topical journal.
- 2) Are there 'crystalline' aspects of the Xe nanoclusters that are adding to the effect here? Does the regular orientation of the clusters contribute to the strong diffraction and as that orientation is reduced, does this also contribute to reduction in scattering signal. See for instance Barty et al, Nature Photonics volume 6, pages 35–40 (2012).
- 3) In Fig. 1, is it a fair comparison to have two pulses (200fs and <1 fs) at ~730 eV and another at 1500 eV? This is hard to understand when β for Xe is 10 higher at ~730 eV than 1500 eV.
- 4) In Fig. 3, the measured cross sections do not seem to be affected much by the shorter pulse durations. All of the cross sections seem to be on the 10^{-25} m² except for the 1500 eV pulse. Would not a 1500 eV pulse of 200 fs show the same effect?

Minor concerns:

- 1) In Fig. 1, the right column states 'scaled to XFEL fluence' perhaps a better label would be 'simulated' or 'expected.'
- 2) In Fig. 2, does the y-axis value have any meaning for the wave-function ψ (i.e. 2 vs 1)? If so this should be explained, otherwise, maybe it should be listed as arbitrary?
- 3) Some of the links in the references do not work (i.e. the DOI links did not open to the correct articles for me)
- 4) The long captions of the figures and multiple sub-figures are hard to digest without significant time and effort (especially figure 2). It might be much clearer to split this figure into at least two other figures.
- 5) In Fig. 3 caption, it reads "increased brightness translates to higher spatial resolution." spatial resolution should be two words.

Reviewer #3

(Remarks to the Author)

The manuscript presents an exciting observation of enhanced diffraction at specific photon energies for ultrashort X-ray pulses. This is practical, as it potentially paves the way to signal enhancement from small, weakly scattering samples, which may not be observed easily, even with the brilliance of an X-ray Free Electron Laser.

In general, the work is significant in the field of XFEL coherent diffraction and (mostly) cites the relevant literature and provides good context. The work relies on a careful calibration of incident intensities and diffracted signals, in a challenging regime with fluctuating pulses, fluctuating particle sizes and fluctuations in the position of the particles in the beam. The careful work shown in the Supplemental Material provides confidence in the overarching results.

One absence in this manuscript is, however, a discussion or outlook about the general practicality of this method for samples of scientific interest. The Summary and Outlook has a few sentences in this direction (page 2, line 163-166), though it would be more useful to the reader to note whether this method may be useful to practically relevant systems, such as core-shell nanoparticles or single biomolecules with metal atoms present (notwithstanding the present challenges in bio-SPI). While these two examples may or may not benefit from the TRs described herein, the paper will be enhanced by providing clearer guidance for a general reader on applicability.

Beyond that, there are just a few minor comments to address:

- In the introduction, a number of XFEL SPI references are cited. These could be more extensive, and cite work from other groups. A cursory literature search will readily reveal a broader cohort of references to cite.
- Page 2, line 23-25 discusses electronic radiation damage during the femtosecond pulse duration. As per the previous comment, there are key groups producing relevant work not cited here. Again, a cursory literature search and a few more, broader citations, will help the reader significantly.

- Page 2, line 45 asserts that 200 fs is the "typical" pulse duration preferred in most ultrafast studies. I'd suggest to be a little more circumspect about this number, as different facilities have different "default" pulse durations. Furthermore, we know that shorter pulse durations (tens of fs) are being used more commonly in present studies.

- Page 3, line 62. Typo: "dimer" to "dimmer".

- Page 5, line 169. It's asserted that < 10 fs pulses have received "...little attention so far in CDI." Again, there is a body of modeling literature exploring this very question which isn't cited here.

- Page 6, figure 1. The label "scaled to fluence" should perhaps better read, "scaled to incident fluence" for clarity for the reader.

- Page 8, figure 3 caption. Typo: "Single exposure...".

- Page 10, Reference 25 - I think "Howellst" is really "Howells". Please check.

- Page 15, page 376 to 384. It would be beneficial to describe the temporal pulse profiles for the reader, either graphically if measured, or at least broadly if not. It is known that these cases potentially each have differing pulse profiles, which can potentially impact results, so best this detail is noted in the paper.

- Page 19, line 489: Typo: "lowerst" to "lowest"

- Page 19, line 491: Typo. "An time of flight..." to "A time of flight..."

- Page 20, Figure 9 caption. Typo: "...was therefore not correct...", to (probably) "...was therefore not corrected..."

In summary, this manuscript is a solid contribution to the field and would benefit the community to be published with these minor amendments.

Version 1:

Reviewer comments:

Reviewer #1

(Remarks to the Author)

The revised version provided some necessary information raised to the original version of the manuscript to be more informative and self-contained. Overall, even not well enough, the revision has fulfilled the inquiry and I may recommend the publication.

However, before the publication, I still strongly recommend the addition of the content described in the supplemental materials as simulation result (with text mostly colored in blue) to the main text to improve the solid standing of the manuscript in overall content, as the suggested implementation is essential to help readers to understand what the authors have done.

Reviewer #2

(Remarks to the Author)

The article "Non-linear enhancement of ultrafast X-ray diffraction through transient resonances" by Kuschel et al. has been substantially edited and has satisfied my concerns from the last review. The impact and importance comes through much more clearly in the current version and more clearly presents this technique as one that might have more general applications, perhaps into time resolved XRD and possibly serial femtosecond crystallography. The additional text and improved figures highlight this clearly and convincingly. I believe after a few more minor changes, it is ready to be published in Nature Communications. I do not need to see the paper again if the following items can be addressed

1. The additional available data is not described in the supplement clearly and the individual figures are not labeled so it is difficult to tell what is what. Perhaps some additional short description can be added to the supplement with file or figure names or short captions can be added to the additional data pdf pages?
2. Caption to Figure 2 ends abruptly with "simulations and its integral is normalized to one" with no punctuation, maybe this is a display error, but it should be fixed.
3. In equation 6 of the supplement, $f_{0,j}(q,C_j)$ is not explicitly defined, maybe it should be?

Reviewer #3

(Remarks to the Author)

The amended work addresses the concerns raised by this referee and represents a useful contribution to the field. In particular, addressing the possible context is appreciated. The expanded section on simulation is also appreciated, and provides at least some insight into the potential choice of parameter space in an applied experiment (though I'd encourage a polish of the Figure 5 caption, particularly to make very clear which parts reference a) and b), and which c) and d)--which

one can work out, though could be better).

We would like to again thank all referees for the constructive and important feedback, and hope that our improved paper is now fully acceptable for publication. The modified and newly added text is highlighted in the manuscript in blue.

REVIEWER COMMENTS

Reviewer #1 (Remarks to the Author):

The manuscript by S.Kuschel, et al. reports new investigation addressing the core-level ionization induced strong radial contraction of the valence orbitals with sensitive dependence on the XFEL pulse duration. Importantly such alteration gives a direct impact on the X-ray scattering cross section of the target element, which can be employed as a cleaver route to enhance diffraction signals. The diffraction signal enhancement is attained from the increase in radial overlap integral to effectively accommodate larger scattering cross section. Overall, the manuscript is well designed by focusing on this point clearly with convincing demonstration combing both experimental data and simulation results. The influence of the XFEL pulse duration on the diffraction signal had been pointed out while ago, which could not receive good attentional without experimental evidence. The presented research provides a clear and impactful advance now to demonstrate it in real experiments. As such, I think the manuscript worth publication in the current journal. I add the following points to be addressed before its publication to make the manuscript scientifically solid.

Thank you for this positive evaluation.

The message becomes unclear if readers want to gain idea about what the sensitive parameter is and how to reach optimum condition. The reason comes from the structure of the manuscript dealing with three parameters of laser fluence, pulse duration, and pulse energy, but without systematic description/investigation. For instance, the pulse duration shorter than AA fs is recommend based on BB analysis. Pulse intensity dependence for each pulse duration needs to be compared for the diffraction signal. And finally how the photon energy, with respect to the absorption edge (or core-level binding energy), shows the dependence on the strength of the diffraction signal? All these make multi-dimensional parameter analysis, which is essential for reaching optimal experimental condition as the manuscript's main message points to. I understand providing multi-dimensional analysis data from experiments is not amenable, but at least theoretical investigation has to be provided systematically.

*This is an important comment that emphasizes the main point of our publication, which is that FEL parameters need to be chosen very careful for each experiment depending on the sample. From the comment of Referee 1 we see that we have not be able to adequately transmit this point so far. **In response to these comments as well from other reviewers, we have dedicated a new subsection in Supplements to the theoretical investigation of the ideal FEL parameters and have added a new Figure 5 which now includes resonance maps from Figure 2 and new, more extensive simulations.***

In addition, we have added the sentence to main text: "Our study demonstrates that optimal conditions for diffraction enhancement can be identified via simplified static resonance maps or more complex

time-resolved simulations as described in supplements.”

1. In Fig.2, please state clearly which one is cartoon picture, and which ones are real simulation results. How was the gray dotted and solid lines are obtained?

To clarify this ambiguity we have added: “The gray lines represent a schematic drawing of the Coulomb potential. The shape of the wave functions is a result of the Hartree-Fock simulations and its integral is normalized to one.”

2. Explicit relation between the scattering cross section and overlap integral needs to be stated in the manuscript.

The equations are now added to the “Simulation” sections in the Supplemental Material, please see equations 5-7.

3. It is also important to state about the quality of the diffraction signals themselves in addition to the signal strength only. The authors should add enough investigation results on this. For instance, any occurrence of distorted diffraction signal with fluence and pulse duration dependence?

This is a crucial aspect, we have checked the quality of the data for all conditions on image-to-image basis. We did not observe distorted diffraction signals for any pulse duration. We have added the original images and our plots of most intense shots for every condition and will post this as an open data set. To also address this issue to the readers, we have added a comment in the subsection “Single exposure images” pointing to the available data.

Reviewer #2 (Remarks to the Author):

The article “Non-linear enhancement of ultrafast X-ray diffraction through transient resonances” by Kuschel et al. presents an interesting application of single pulse x-ray free electron laser radiation to study Xe nanoclusters. The authors illuminated nanoclusters with single XFEL pulses of various pulse lengths and show compelling evidence that transient resonances induced by the highly ionized states greatly increase the scattering efficiency. The results seem valid and compelling, the presentation is clear and concise. However, the current state of the paper does not seem sufficiently compelling to merit publication in Nature Communications unless the following questions could be addressed.

We are grateful for such a thorough review recognizing the quality of our work.

Major concerns/questions:

1) **Does the current technique have any applicability outside of Xe nanoclusters?** Can this transient resonance impact, for instance, biological cell imaging or serial femtosecond crystallography? What evidence is there of this being applicable to other areas of XFEL or x-ray science beyond Xe nanoclusters? This argument to me seems to greatly affect whether the impact of this work is sufficiently broad for a wider audience such as Nature Communications readers, or whether this work should appear in a more focused topical journal.

We agree that the generality of transient resonances is a fundamentally important point and Xe nanoparticles serve here as a model system as they generate a stronger X-ray diffraction signal compared to organic samples. As a further benefit, the electronic structure and response of Xe to intense X-ray radiation has been well characterized in the literature including transient absorption increase in Xe ions at 1500 eV (Rudek 2012, Nat. Phot.).

The principle of transient resonances is not exclusive to Xe. The increased overlap between orbitals in short-lived ions can be expected in any multi-shell element. Transient resonances have been observed as enhanced absorption in other elements such as neon (ref 47) and oxygen (ref 38) . In follow-up experiments we have experimentally observed indications for the enhancement of coherent diffraction through transient resonances in Ne clusters in the soft X-ray regime and even for Bragg diffraction in the hard X-ray regime. Based on these newer data, our confidence that transient resonances will play a strong role in all future X-ray diffraction and images experiments has only grown and thus we believe that the results on Xe deserve broad dissemination.

Following this comment as well as from the next referee, we have changed the outlook and included some general application examples. We now state that: “.. Transient resonances can be expected in any multi-shell element. In the soft X-ray regime transient resonances could be combined with super-resolution algorithms to achieve high material contrast from light elements, such as carbon, oxygen and nitrogen, and high spatial resolution down to single molecule level. In principle, TRs should also exist in the hard X-ray regime which offers a direct route to Ångstrom resolution in single shot X-ray imaging of fragile or short-lived states of matter. This would open a plethora of applications such as

observation of non-equilibrium processes under ambient conditions at the single nanoparticle level with sub-fs temporal precision. Such capabilities would benefit our understanding of photochemistry, catalysis, and material studies.”

2) Are there ‘crystalline’ aspects of the Xe nanoclusters that are adding to the effect here? Does the regular orientation of the clusters contribute to the strong diffraction and as that orientation is reduced, does this also contribute to reduction in scattering signal. See for instance Barty et al, Nature Photonics volume 6, pages35–40 (2012).

Thank you for raising this interesting aspect. The orientation is an important factor of the intensity and shape of Bragg peaks. Our images are not sensitive to the crystalline structure because Xe Bragg peaks appear at much higher q -regions, which cannot be reached with soft X-ray wavelengths (see Ferguson 2018, Science Advances for a hard X-ray reference with Xe nanoparticle Bragg peaks appearing on the detector). As pointed out by Barty et al., the brightness of a Bragg peak depends on the crystal orientation, atomic order, scatterer density and the scattering cross-section per scatterer. In the Barty et al. experiment, the atomic disorder has “gated” the Bragg peak intensity for longer pulses.

In our experiment, we have covered lower q -regions, which are sensitive exclusively to the scatterer density and the scattering cross-section per scatterer, but not to the atomic order. For example, we cannot distinguish amorphous from crystalline nanoparticles based on the current diffraction images.

3) In Fig. 1, it it a fair comparison to have two pulses (200fs and <1 fs) at ~730 eV and another at 1500 eV? This is hard to understand when beta for Xe is 10 higher at ~730 eV than 1500 eV.

This is a valid discussion point, in the ideal case we should have recorded a sub-fs FEL pulse data set at 1500 eV. Unfortunately, at the time of the experiment XLEAP sub-fs pulses were only available up to 900 eV photon energy and it was impossible to record this data point. As mentioned previously, we have recorded the 1500 eV set based on the results from the Rudek et. al. Nat. Phot. 2012 study.

Despite the missing reference at 1500 eV, the overall recorded data set provides solid evidence for the existence and the enhancement of ultrafast diffraction images through transient resonances for short FEL pulses.

We have analyzed the data from two perspectives. First, we compare the brightest images from all pulse durations to the linear semiclassical model of diffraction. The images reflect the imaginary (beta) and the real part (delta) of the refractive index. The comparison between data sets recorded at 730 eV (near the M-edge) and 1500 eV (far above the M-edge) show clear deviations from the linear model. In the first figure, the images are compared to simulated images based on literature values which disregard the presence of transient resonances and structural damage. Images recorded with short pulses are brighter than predicted by the linear model, which indicates enhancement of scattering through transient resonances. However, this perspective is

not sufficient to understand the deviations from the linear model. Here the second perspective, which separates beta and the scattering response, is needed.

In this next step, we compare the unfiltered data set of all recorded images where beta is reflected through fluorescence, which is recorded simultaneously to each diffraction image (see Figure 9 and the sub-section "Signature of transient resonances in the fluorescence yield"). Here, the presence of transient resonances is mirrored in the absorption efficiency. As pointed out by the referee, the beta (imaginary) part of the refractive index at 1500 eV (beta=18) should be significantly smaller than at 730 eV (beta=38) according to literature (henke.lbl.gov). Thus, one would expect a high rate of fluorescence per atom for the 730 eV data points. In contrast, we observe that fluorescence per atom rises to extreme values at the 1500 eV data point. The same data point also exhibits the highest scattering cross section, which is consistent with the appearance of transient resonances without the negative impact from structural damage.

Finally, our data is supported by our simulation (displayed in Figure 3 (a)-(c), solid lines), which reconstructs the overall signatures of transient resonances.

4) In Fig. 3, the measured cross sections do not seem to be affected much by the shorter pulse durations. All of the cross sections seem to be on the 10^{-25} m^2 except for the 1500 eV pulse. Would not a 1500 eV pulse of 200 fs show the same effect?

Our presentation of the data was misleading, we are grateful to the referee for pointing out this important aspect. The measured cross sections are affected dramatically by transient resonances in all pulse duration settings, which was hidden in the logarithmic scale of our plots of Figure 3 (a) and (c), please see new Figure 3 with linear scale graphs. Our data reflects that the pulse duration plays a critical role in the scattering efficiency when the FEL exposure intensity is increased. Thus, the cross sections deviate greatly from 10^{-25} m^2 for all pulse durations at certain photon energies.

For moderate FEL experiment intensities (<1-2 absorbed photons per atom), transient resonances increase the scattering cross section for 200 fs pulses by factors of up to 6-8 (see Figure 3 (b), violet dots, 730 eV data point). In the case of sub-fs pulses, we have observed an increase of the scattering cross section beyond the literature values by up to factor of 3-4 (see Figure 3 (c), yellow dots, 730 eV data points). For higher FEL exposure intensities, all 200 fs data points near resonances exhibit a significantly decreased scattering efficiency (see Figure 3 (b), dark blue dots, all data points).

It is true that one or two more data points would have been a wonderful measurement, but unfortunately FEL beam time is limited. However, our full data set in combination with agreeing simulations paints a consistent picture about the interplay of transient resonances and the FEL pulse duration.

First, the 200 fs data set in combination with the simulation demonstrates in a consistent manner that transient ion resonances can lead to a rapid disintegration of the sample, which decreases the scattering cross section significantly and degrades the quality of the images (see Figure 3, panel (b), blue dots). This is consistent with prior observations in imaging experiments with approx. 200 fs pulses

of sugar nanospheres near the oxygen edge (see Ho et al., 2020 Nature Communication).

Second, the interplay between absorption, resonances and pulse duration is displayed in Figure 9 in the unfiltered data set of images recorded at photon energies affected by transient resonances. In the right panel, the correlation between absorption (fluorescence) and scattering per atom are plotted for individual images (single dots) for the photon energy 730 eV and 200 fs pulse duration (dark blue and violet dots). Within the data set, the fluorescence increases with the FEL exposure intensity. The data points near the highest fluorescence/FEL power density saturate in scattering. That means that the ions continue to absorb photons, but the scattering signal does not increase. This behavior is a signature of a rapid ion density decrease, which can be expected during a hydrodynamics expansion (similarly as observed in previous experiments Gorkhover et al, Nature Photonics). This experimental finding is supported by our simulation, which demonstrates that the ionic density will undergo a significant dilution already on the time scale of tens of femtoseconds (see Figure 4).

In contrast, the 1500 eV data point combines highly enhanced fluorescence as well as scattering for 5-10 fs pulse durations. Here, the fluorescence efficiency is almost three orders of magnitude higher than the literature value for the M-edge (the grey dotted line, assuming only one absorption event per atom), which indicates an extreme number of absorption events per atom. Despite such increased absorption, there is no saturation in scattering because the pulses are shorter than the expected ion structure expansion.

Minor concerns:

1) In Fig. 1, the right columns states 'scaled to XFEL fluence' perhaps a better label would be 'simulated' or 'expected.'

We have changed it to "linear model prediction".

2) In Fig. 2, does the y-axis value have any meaning for the wave-function ψ (I.e 2 vs 1)? If so this should be explained, otherwise, maybe it should be listed as arbitrary?

We have added: Note the y-axis in (a) and (b) are in arbitrary units.

3) Some of the links in the references do not work (i.e. the DOI links did not open to the correct articles for me)

Thank you for thorough reading, this should be fixed now.

4) The long captions of the figures and multiple sub-figures are hard to digest without significant time and effort (especially figure 2). It might be much clearer to split this figure into at least two other figures.

We have followed this advice and transferred parts of figure 2 to Supplements in combination with

a more thorough discussion about our simulation responding to the first referee.

5) In Fig. 3 caption, it reads “increased brightness translates to higher spatial resolution.” spatial resolution should be two words.

Done.

Reviewer #3 (Remarks to the Author):

The manuscript presents an exciting observation of enhanced diffraction at specific photon energies for ultrashort X-ray pulses. This is practical, as it potentially paves the way to signal enhancement from small, weakly scattering samples, which may not be observed easily, even with the brilliance of an X-ray Free Electron Laser.

In general, the work is significant in the field of XFEL coherent diffraction and (mostly) cites the relevant literature and provides good context. The work relies on a careful calibration of incident intensities and diffracted signals, in a challenging regime with fluctuating pulses, fluctuating particle sizes and fluctuations in the position of the particles in the beam. The careful work shown in the Supplemental Material provides confidence in the overarching results.

Thank you for this thoughtful review.

One absence in this manuscript is, however, a discussion or outlook about the general practicality of this method for samples of scientific interest. The Summary and Outlook has a few sentences in this direction (page 2, line 163-166), though it would be more useful to the reader to note whether this method may be useful to practically relevant systems, such as core-shell nanoparticles or single biomolecules with metal atoms present (notwithstanding the present challenges in bio-SPI). While these two examples may or may not benefit from the TRs described herein, the paper will be enhanced by providing clearer guidance for a general reader on applicability.

We agree that the general aspect and importance of this effect was not properly reflected in the initial version, as also pointed out by other referees. Our newer data indeed indicates that transient resonances do not only play a role in imaging but also Bragg scattering. Therefore transient resonances can turn out to be a novel approach to solve the high-intensity x-ray investigation limitations.

In response to this comment we have added additional discussion in the outlook: “Transient resonances can be expected in any multi-shell element. In the soft X-ray regime transient resonances could be combined with super-resolution algorithms to achieve high material contrast from light elements, such as carbon, oxygen and nitrogen, and high spatial resolution down to single molecule level. In principle, TRs should also exist in the hard X-ray regime which offers a direct route to Ångström resolution in single shot X-ray imaging of fragile or short-lived states of matter. This would open a plethora of applications such as observation of non-equilibrium processes under ambient conditions at the single nanoparticle level with sub-fs temporal precision. Such capabilities would benefit our understanding of photochemistry, catalysis, and material studies. Beyond that, TRs are

element specific and thus could increase material contrast in X-ray chemical femtosecond crystallography and potentially eliminate the need of heavy stain atoms in time-resolved multi-wavelength anomalous crystallography studies. The optimal conditions for diffraction enhancement through transient resonances can be found through simulations such as demonstrated in supplements in Figure 5. In principle, transient resonances can be excited by an optical pre-pulse and significantly increase signal-to-noise ratio in state-of-the-art ultrafast table-top XUV imaging setups.

Beyond that, there are just a few minor comments to address:

- In the introduction, a number of XFEL SPI references are cited. These could be more extensive, and cite work from other groups. A cursory literature search will readily reveal a broader cohort of references to cite.

We have added 4 more references (Chapman2011, Seibert2011, VanDerSchot2015, Hantke2014) which refer to the biological studies.

- Page 2, line 23-25 discusses electronic radiation damage during the femtosecond pulse duration. As per the previous comment, there are key groups producing relevant work not cited here. Again, a cursory literature search and a few more, broader citations, will help the reader significantly.

We have added 3 more references (Jurek2008t, Lorenz2012, Son2011) which refer to damage in SPI.

- Page 2, line 45 asserts that 200 fs is the "typical" pulse duration preferred in most ultrafast studies. I'd suggest to be a little more circumspect about this number, as different facilities have different "default" pulse durations. Furthermore, we know that shorter pulse durations (tens of fs) are being used more commonly in present studies.

Our wording was not very precise and we thank the referee for pointing this out. It is true that in soft X-ray SPI experiments the pulse duration mostly ranges between 80 to 200 fs. We have changed the sentence to "> 100 fs as a pulse duration found in many soft X-ray CDI studies". Our main reference for the pulse duration was Ho 2020, Nat. Comm., which was performed in preparation to a biological CDI experiment with commonly used FEL settings. In this publication, the impact of transient resonances was negative on the image due to long pulse duration and we have successfully reproduced this observation for Xe nanoparticles.

- Page 3, line 62. Typo: "dimer" to "dimmer". *Done*

- Page 5, line 169. It's asserted that < 10 fs pulses have received "...little attention so far in CDI." Again, there is a body of modeling literature exploring this very question which isn't cited here.

We have changed the sentence to "...little attention so far in experimental CDI..". We have added more theoretical studies which explored pulse duration dependence into the intro as suggested by the referee in the previous section.

- Page 6, figure 1. The label "scaled to fluence" should perhaps better read, "scaled to incident

fluence" for clarity for the reader. *Done*

- Page 8, figure 3 caption. Typo: "Single exposure...". *Done*

- Page 10, Reference 25 - I think "Howellst" is really "Howells". Please check. *Done*

- Page 15, page 376 to 384. It would be beneficial to describe the temporal pulse profiles for the reader, either graphically if measured, or at least broadly if not. It is known that these cases potentially each have differing pulse profiles, which can potentially impact results, so best this detail is noted in the paper.

We have added: "The temporal profiles of 5-200 fs FEL pulses are not precisely known, the pulse duration is based on indirect electron bunch measurements." to the experiment description in the supplements. The pulse duration measurement of XLEAP is cited on page 15.

- Page 19, line 489: Typo: "lowerst" to "lowest" *Done*

- Page 19, line 491: Typo. "An time of flight..." to "A time of flight..." *Done*

- Page 20, Figure 9 caption. Typo: "...was therefore not correct...", to (probably) "...was therefore not corrected..."

In summary, this manuscript is a solid contribution to the field and would benefit the community to be published with these minor amendments

Thank you for careful reading!

REVIEWERS' COMMENTS

We would like to thank all the reviewers for the thorough reviews our manuscript. We sincerely appreciate the time and effort you dedicated to providing constructive feedback, which has significantly improved the quality of our work.

Reviewer #1 (Remarks to the Author):

The revised version provided some necessary information raised to the original version of the manuscript to be more informative and self-contained. Overall, even not well enough, the revision has fulfilled the inquiry and I may recommend the publication.

However, before the publication, I still strongly recommend the addition of the content described in the supplemental materials as simulation result (with text mostly colored in blue) to the main text to improve the solid standing of the manuscript in overall content, as the suggested implementation is essential to help readers to understand what the authors have done.

The simulation results are now included in the main text. Thank you for your valuable feedback!

Reviewer #2 (Remarks to the Author):

The article “Non-linear enhancement of ultrafast X-ray diffraction through transient resonances” by Kuschel et al. has been substantially edited and has satisfied my concerns from the last review. The impact and importance comes through much more clearly in the current version and more clearly presents this technique as one that might have more general applications, perhaps into time resolved XRD and possibly serial femtosecond crystallography. The additional text and improved figures highlight this clearly and convincingly. I believe after a few more minor changes, it is ready to be published in Nature Communications. I do not need to see the paper again if the following items can be addressed

1. The additional available data is not described in the supplement clearly and the individual figures are not labeled so it is difficult to tell what is what. Perhaps some additional short description can be added to the supplement with file or figure names or short captions can be added to the additional data pdf pages?

Done

2. Caption to Figure 2 ends abruptly with “simulations and its integral is normalized to one” with no punctuation, maybe this is a display error, but it should be fixed.

Done

3. In equation 6 of the supplement, $f_{0,j}(q,C_j)$ is not explicitly defined, maybe it should be?

Done.

We greatly appreciate your suggestions!

Reviewer #3 (Remarks to the Author):

The amended work addresses the concerns raised by this referee and represents a useful contribution to the field. In particular, addressing the possible context is appreciated. The expanded section on simulation is also appreciated, and provides at least some insight into the potential choice of parameter space in an applied experiment (though I'd encourage a polish of the Figure 5 caption, particularly to make very clear which parts reference a) and b), and which c) and d)--which one can work out, though could be better).

The former Figure 5 is now clearly separated in two parts in the caption text. Thank you for all your constructive input!